# Minimally-Invasive Secondary Cytoreduction in Recurrent Ovarian Cancer

**DOI:** 10.3390/cancers15194769

**Published:** 2023-09-28

**Authors:** Camilla Certelli, Silvio Andrea Russo, Luca Palmieri, Aniello Foresta, Luigi Pedone Anchora, Virginia Vargiu, Francesco Santullo, Anna Fagotti, Giovanni Scambia, Valerio Gallotta

**Affiliations:** 1Gynecologic Oncology Unit, Department of Woman and Child Health and Public Health, Fondazione Policlinico Universitario Agostino Gemelli IRCCS, 00168 Rome, Italy; 2Institute of Obstetrics and Gynecology, Università Cattolica del Sacro Cuore, 00168 Rome, Italy; 3Surgical Unit of Peritoneum and Retroperitoneum, Fondazione Policlinico Universitario A. Gemelli IRCCS, 00168 Rome, Italy

**Keywords:** recurrent ovarian cancer, secondary cytoreductive surgery, minimally-invasive surgery, robotic surgery, laparoscopic surgery

## Abstract

**Simple Summary:**

Ovarian cancer (OC) represents one of the most lethal cancers in women, with most cases diagnosed at an advanced stage. Recurrence occurs in around 70% of women within 5 years of diagnosis. In this era, in which maintenance therapy with PARP-inhibitors is the standard of care, most recurrent OC (ROC) patients are platinum-sensitive, and the choice of treatment becomes crucial. Randomized clinical trials investigated the role of surgery plus chemotherapy in the treatment of ROC, underlying an advantage in terms of progression-free survival and overall survival compared with chemotherapy alone. New recommendations concluded that platinum-sensitive OC patients should be assessed for eligibility for secondary cytoreductive surgery (SCS) in a gynecological oncology center. In selected cases, a minimally-invasive approach can be used. This narrative review is focused on minimally-invasive SCS and on the wide range of elements that must be considered in patient selection.

**Abstract:**

The role of secondary cytoreductive surgery (SCS) in the treatment of recurrent ovarian cancer (ROC) has been widely increased in recent years, especially in trying to improve the quality of life of these patients by utilising a minimally-invasive (MI) approach. However, surgery in previously-treated patients may be challenging, and patient selection and surgical planning are crucial. Unfortunately, at the moment, validated criteria to select patients for MI-SCS are not reported, and no predictors of its feasibility are currently available, probably due to the vast heterogeneity of recurrence patterns. The aim of this narrative review is to describe the role of secondary cytoreductive surgery and, in particular, minimally-invasive procedures, in ROC, analyzing patient selection, outcomes, criticisms, and future perspectives.

## 1. Introduction 

Ovarian cancer (OC) represents one of the most lethal cancers in women, with a rate of new cases of 10.6 per 100,000 women per year, a death rate of 6.3 per 100,000 women per year [1], and about 80% of cases in advanced FIGO stage at the time of diagnosis [2].

Recurrence occurs in around 70% of women within 5 years of diagnosis [3]; therefore, the management of these patients represents a significant clinical challenge. The introduction of bevacizumab and PARP-inhibitors (PARP-i) as maintenance therapy in first-line treatment opened novel scenarios in the management of recurrent ovarian cancer (ROC). Recent data, in fact, showed that more than 80% of patients experienced a recurrence after 6 months from the last cycle of platinum [4,5,6,7]. The use of bevacizumab improved progression-free survival (PFS) in women with OC (PFS of 20.3 months with standard therapy vs 21.8 months with standard therapy plus bevacizumab (hazard ratio for progression or death with bevacizumab added, 0.81; 95% confidence interval, 0.70 to 0.94; *p* = 0.004)) [7]. The use of maintenance therapy with olaparib provided a substantial benefit with regard to PFS among women with newly-diagnosed advanced OC and a BRCA1/2 mutation, with a 70% lower risk of disease progression or death with olaparib than with placebo [6]. In PAOLA-1, in patients with advanced ovarian cancer receiving first-line standard therapy including bevacizumab, the addition of maintenance olaparib provided a significant PFS benefit, which was substantial in patients with homologous-recombination deficiency (HRD)-positive tumors, including those without a BRCA mutation [4]. Furthermore, those who received niraparib had significantly longer PFS than those who received placebo (PFS was 13.8 months and 8.2 months, respectively (hazard ratio, 0.62; 95% CI, 0.50 to 0.76; *p* < 0.001)), regardless of the presence or absence of HRD [5].

Therefore, in a population of platinum-sensitive ROC patients, the choice of treatment becomes crucial. The standard treatment of ROC patients usually includes intravenous chemotherapy chosen based on the platinum sensitivity. Since the first suggestion by Berek et al. [8] in 1984, the potential role of secondary cytoreductive surgery (SCS) in recurrent ovarian cancer has been increasingly considered, leading to worldwide acceptance of this treatment option, despite the retrospective design and heterogeneity of most of the studies [8,9,10,11,12,13,14,15,16,17,18,19]. Afterwards, randomized clinical trials investigated the role of surgery plus chemotherapy in the treatment of ROC, underlying an advantage in terms of PFS and overall survival (OS) compared with chemotherapy alone [20,21,22]. The first randomized controlled trial (RCT), published in 2019 comparing SCS followed by chemotherapy vs chemotherapy alone in the treatment of platinum-sensitive ROC (GOG-0213 trial), showed that SCS followed by chemotherapy did not result in longer OS than chemotherapy alone (median OS of 50.6 and 64.7 months, respectively; HR = 1.29, *p* = 0.08) [20]. However, in the overall study population, 84% of patients received bevacizumab, and it may have masked an incremental benefit from surgery. The median OS for the entire trial cohort was nearly three times longer than expected when the trial was designed, diluting the treatment effect by enabling a higher likelihood of intervening treatment. As reported by the authors, the reasons for these results are unknown, but they could be related to the introduction of target therapies, such as PARP-i, in recent years. Furthermore, in this study, BRCA mutational status or the use of PARP-i therapy that may have improved PFS were not investigated. In 2021, the publication of another two RCTs clarified the role of SCS in recurrent ovarian cancer. In the DESKTOP-III trial, the treatment of women with platinum-sensitive ROC using SCS resulted in longer OS than chemotherapy alone (median OS of 53.7 months with and 46.0 months without surgery (HR = 0.75, 95% CI: 0.58–0.96, *p* value = 0.02), without perioperative mortality within 30 days after surgery and no differences in quality of life between the two groups [21]. The SOC-1 trial showed a longer PFS in patients treated with SCS followed by chemotherapy than chemotherapy alone (median PFS of 17.4 months in the surgery group and 11.9 months in the no surgery group (HR = 0.58, 95% CI: 0.45–0.74; *p* < 0.0001)), although data on OS are not yet mature [22].

Therefore, new oncological strategies in OC prolonged the treatment time, driving our efforts in chronicity of the disease. In this context, SCS plays a crucial role as a treatment to remove tumor clones that are probably resistant to chemotherapy, especially in oligometastatic disease. The single recurrence rate in OC, in fact, is reported to be up to 30% [23] (and some recent studies showed continuation of PARP-i after locally treated progression in oligometastatic ovarian cancer recurrences, with the hypothesis that local therapy destroys and reduces potential tumor clones developing secondary resistances to PARP-i [24,25]).

In recent years, minimally-invasive surgery (MIS) techniques have been increasingly used in gynecological oncology practice because of the several benefits reported with respect to the open approach [26,27,28,29,30,31,32]. In particular, MIS has been successfully employed in selected ROC patients presenting a lower incidence of morbidities, apparently without compromising their survival [33,34,35].

However, the technical difficulties of surgery in a previously-treated patient (both with surgery and chemotherapy) may be among the current challenges of SCS, and particularly MI-SCS. Furthermore, since, as in the primary debulking surgery, the most important prognostic factor for SCS is complete cytoreduction, patient selection and surgical planning play a crucial role in the surgical treatment of ROC.

Recent recommendations suggest considering for SCS all platinum-sensitive ROC patients [36]. However, at the moment, validated criteria to select patients for MI-SCS are not reported, and no predictors of its feasibility are currently available, probably due to the vast heterogeneity of recurrence patterns.

This review focuses on the role of MI-SCS, analyzing the aspects of patient selection and outcomes reported in literature, along with their limits and future perspectives.

## 2. Patient Selection

The most relevant data emerging from the results of the RCTs on SCS is that patients with complete gross resection have the most favorable outcome. The GOG 0213 and DESKTOP-III trials reported a complete cytoreduction rate of 67% and 75.5%, respectively, underlying that only this group of patients really benefit from SCS [20,21]. For this reason, patient selection to identify optimal candidates for SCS is crucial, especially in the choice of MIS. The AGO score, used in the DESKTOP trials, considers patient performance status (PFS = 0), the presence of ascites (<500 cc), and the residual disease at first surgery (RT = 0) [21]. Similarly, the TIAN score, used in the SOC-1 trial, in which FIGO stage, residual disease at primary diagnosis, PFI, ECOG, and CA125 levels, and ascites are considered, with a complete cytoreduction in the low-risk group of 53.4% [22]. However, a very high rate of false-negative results for both algorithms (68.5% vs. 55.6%) was also observed [37]. So far, there is no guidance defining the ideal candidate for MI-SCS, and no predictors of its feasibility are currently available. Therefore, the choice regarding the minimally-invasive approach to SCS is left to the surgeon’s discretion, but there are different points to consider.

Age, clinical conditions, history (previous surgeries or radiotherapies), BMI, and pattern of recurrence may affect the choice of the best surgical approach. Thanks to a multidisciplinary collaboration, there will be a greater synergy for determining surgical indications and approaches in this particular setting of patients. The use of MIS is particularly interesting, as the reduced surgical trauma can potentially lead to a reduction of postoperative complications. However, the choice of surgical approach depends not only on surgeon preference, but also on clinical and intraoperative findings: for example, the influence of medical comorbidities, the patient’s ability to tolerate steep positions (Trendelenburg), and pneumoperitoneum. For this reason, anesthesiologic evaluation plays an important role in the planning of surgery [38,39].

The pattern of relapse influences prognosis. Petrillo et al. showed relevant survival differences according to the anatomic site of recurrent disease: lymph-nodal recurrences have the best outcome, followed by peritoneal and parenchymal recurrences, probably due to the still poorly-known biological differences existing between the different patterns of recurrence. Furthermore, they confirmed the favorable prognosis of ovarian cancer patients with localized relapse, reporting a median PFI and PRS of 13 and 40 months, respectively. Interestingly, they reported that around 10% of patients with peritoneal or lymph node recurrences who underwent SCS were alive and free from disease after more than 5 years from the diagnosis of localized relapse, thus suggesting that SCS in this specific clinical setting can ensure, in a non-negligible percentage of cases, long-term healing from cancer [23]. In a recent analysis of more than 276 patients, comparing laparotomic- and MI-SCS, predictors of successful a minimally-invasive approach were analyzed through univariate and multivariate analysis. Neoadjuvant chemotherapy at primary diagnosis, peritoneal and lymph-nodal pattern of recurrence, and single and oligometastatic disease at preoperative PET/CT scan were significantly correlated with MI-SCS feasibility. In the multivariate analysis, single disease (OR = 4.17, 95% CI: 1.83–9.53, *p* < 0.001) and oligometastatic disease at recurrence (OR = 3.46, 95% CI: 1.48–8.07, *p* = 0.004), neo-adjuvant chemotherapy at primary diagnosis (OR = 2.46, 95% CI: 1.27–4.78, *p* = 0.007), and lymph-nodal site (OR = 2.67, 95% CI: 1.09–6.53, *p* = 0.031) were confirmed to be independent predictors of MI- SCS [40]. They showed, for the first time, a surprisingly higher rate of neoadjuvant chemotherapy and interval debulking surgery (IDS) at first diagnosis (*p* = 0.030) in the MIS group. It is likely that patients undergoing neoadjuvant chemotherapy may receive a less complex surgery at IDS, with consequently fewer adhesions and/or a different pattern of recurrence, potentially favorable to a minimally-invasive approach. Considering that the IDS population has significantly increased in the last years, representing up to 42.2% of primary ovarian cancers [41], the use of MI-SCS could increase accordingly in the coming years. Less recent studies, instead, reported that neoadjuvant chemotherapy plus IDS negatively influenced the pattern (more carcinomatosis at recurrence in the IDS group) and timing of recurrence (3 year post-relapse survival: primary debulking surgery (PDS) 58% vs IDS 18%) [42]. Bizzarri at al. showed no differences in post-recurrence survival outcomes of ROC patients treated by IDS or PDS (median 21 vs 21 months, *p* = 0.684) [43]. Even in this case, the different results may be affected by the introduction of maintenance therapies with PARP-i.

Considering the increasing complexity of ROC management, the detection of recurrence and its distribution is crucial to decide the best therapeutic option. The precise description of recurrence sites, the potential involvement of adjacent organs, and possible anatomical variants may affect the surgical planning, considering possible technical difficulties and potential complications. In a meta-analysis of 34 studies, PET/CT showed better results than CT or MRI in detecting ROC [44]. In a recent study, Nunes and colleagues evaluated the role of PET/CT in predicting no residual disease after SCS in a cohort of patients with platinum-sensitive recurrent ovarian carcinoma. They showed that, when PET/CT detected uptake in ≤2 sites, 92.3% of cases had a complete SCS, with an accuracy of 81.4% [45]. However, CT is usually the technique of choice in the follow-up and surgical planning of patients with OC because of the reproducibility of imaging for comparison and the excellent spatial resolution for the study of anatomy and recurrent relationships with adjacent structures. Furthermore, imaging may have a role even during surgery, providing a guide in the precise detection of the site of recurrence. Mascilini et al. showed how, in about 25% of cases, intraoperative ultrasound helps in identifying single relapse and in achieving complete cytoreduction by MIS, thus avoiding conversion to laparotomy [46]. The vast application of intraoperative ultrasound offers a precious contribution in the complete gross resection of recurrent disease.

Unfortunately, preoperative imaging is less accurate in the detection of small foci of the recurrent tumor and miliary peritoneal involvement, with lower sensitivity than that of laparoscopy or surgical exploration. In a series by Fanfani et al., nearly 20% of patients with a negative AGO score achieved successful complete secondary cytoreduction after PET–CT and laparoscopic evaluation, and almost one of three positive-AGO score patients would be submitted to an unnecessary explorative laparotomy [47]. The combination of FDG-PET/CT and LPS has, in fact, had a significant effect on the multimodal approach to the patients with ROC. Such techniques should be considered complementary because of the potential of each one to identify a different setting of the disease [48]; laparoscopy still seems to be the gold standard in the final decision-making process. Laparoscopy, in fact, has a favorable impact in terms of better visualization and improvement of adhesiolysis procedures, reducing the risk of an unnecessary exploratory laparotomy.

## 3. The Minimally-Invasive Approach

In recent years, MIS has been increasingly expanding its indications, from early to advanced disease [26,27,28,29,30,49,50,51,52,53,54], and even in recurrent disease. The reported advantages of MIS with respect to the open group include less blood loss and a shorter hospital stay, which is also a factor influencing earlier postoperative recovery, and consequently the possibility of an adequate onset of adjuvant chemotherapy. Furthermore, MIS seems to be not inferior to laparotomy in terms of oncological outcomes. The rate of complete gross resection by MI-SCS, in fact, is consistent across studies, ranging from 79% to 98%, with no differences reported between MIS and laparotomy [15,33,35,40,55,56,57].

Even robotic surgery plays a role in selected cases of ROC undergoing SCS, providing its advantages in terms of 360° movements, tremor filtering, stable 3D vision, and an ergonomic position of the surgeon [58].

However, there are some criticisms to consider: few papers compared MIS and laparotomy in the treatment of ROC, and most of the studies reported only series with a small number of selected cases of MI-SCS. Since all the studies are retrospective, in which the choice of the approach depends on the surgeon, and in recurrent disease, as we explained above, there are many variables to take into account, making it difficult to create a homogeneous data evaluation; therefore, it may be difficult to draw conclusions. In fact, groups are often not balanced in terms of type of recurrences, with a higher rate of extensive procedures in the open group, and there is little information about post-surgery therapies, which may have impacted the outcomes. Furthermore, MIS requires a high level of expertise and skill, especially in this group of patients, and should be performed in high-volume oncological centers with adequate experience in advanced surgical procedures [59].

Furthermore, the same concerns about the application of MIS in early-stage OC may be argued in this setting, such as port-site metastasis and tumor spillage. As far as we know, current data about the rate of port-site metastasis in MI-SCS are missing. In early-stage OC, the reported port-site metastasis rate is very low (less than 1%) [26,52], and in oligometastatic recurrent disease, especially parenchymal or lymph node, without ascites and carcinosis, we may assume the risk is maintained at a low level. However, the use of some precautions, such as the use of endobags, reduced surgical specimen manipulations, and controlled exsufflation, should be always highly encouraged.

In 2013 Magrina et al. analyzed 52 selected patients with ROC undergoing SCS by laparoscopy, laparotomy, or robotics. They showed no significant differences in intra- or postoperative complications, OS, or PFS relative to the number of procedures, type of procedures, or by surgical approach, and there were no differences in operating times, intra- or postoperative complications, or the rate of complete debulking. Statistically significant differences were noted in decreased blood loss and length of hospital stay for the robotic and laparoscopy groups [35]. Fagotti et al. compared MIS and laparotomy in 22 patients with ROC who underwent SCS plus HIPEC, showing an advantage of MIS in terms of intraoperative and postoperative outcomes, despite a similar surgical complexity between the two groups [55].

Eriksson et al. showed their results of 170 ROC patients treated by SCS with the three different approaches, concluding that MIS is associated with favorable perioperative outcomes and similar oncologic outcomes compared to laparotomy [33].

A more recent study compared MIS and laparotomy in the treatment of ROC. In their series, complete gross resection was achieved in 94.9% patients, with a statistically higher rate of complete gross resection in the MI group than in the laparotomic group, probably related to the difference in surgical complexity score, which was significantly higher in the open than in the MI group. Although early post-operative complications were significantly higher in the open group, they showed no differences in terms of oncological outcomes, reporting a 3 year post-recurrence survival rate of 76% and 84.1% in MIS and open groups, respectively. Despite the high number of patients (276), the authors reported a near 10% difference in post-recurrence survival in favor of the open group, highlighting that further analysis in more extensive series are needed to draw conclusions [40].

Given the balance between survival benefit and surgery-related morbidity during the maximum cytoreductive surgical effort, it is essential to establish the optimal selection criteria for identifying appropriate candidates who will benefit from surgery without worsening quality of life [60].

Lymph node recurrence accounts for 12% to 37% of OC relapses [61,62,63,64,65,66,67,68,69], and it is considered more suitable for selected cases to take advantage of surgical rather than medical treatment, because of the low growth rate, relatively lower chemosensitivity, and more indolent behavior compared to parenchymal and peritoneal disease [23].

Since a prospective, randomized study comparing the MIS and open approaches would be very difficult to achieve due to the rarity of this pattern of disease, we can refer only to the available retrospective, small-sample series, thus precluding the possibility of obtaining robust data relative to prognostic factors. Gallotta et al., interestingly, showed no differences in PFS according to surgical approach or extent of lymphadenectomy (systematic versus bulky node resection) in 85 patients who underwent salvage lymphadenectomy for ROC, reporting a complete gross resection exceeding 90%. Only PFI duration > 12 months, and the presence of 3 or less metastatic lymph nodes, were shown to keep their favorable, independent prognostic value in multivariate analysis [70]. Despite surgical management, MIS could be expected to represent a challenging task in the salvage setting, though it may have a favorable impact in terms of better visualization and improvement of adhesiolysis procedures (Figure 1).

A previous study reported successful complete eradication of lymph node relapse in all cases, with acceptable surgical outcomes and morbidities [71], comparable to those reported for secondary lymphadenectomy with open approach. Loverro et al. reported a case of OC superficial celiac isolated nodal recurrence, successfully removed through a MI approach (Figure 2) [72].

The prognostic role of hepatoceliac lymph node involvement has been investigated in primary cytoreductive surgery, representing a marker of disease severity associated with the worst oncologic outcome [73]. However, lymphadenectomy in recurrent disease may be challenging, and the surgeon should be able to prevent and manage severe vascular complications both intraoperatively and postoperatively, and a collaboration with the vascular team is needed. Considering the wide range of sites of recurrence, management of these kind of patients in a multidisciplinary context is mandatory. In a work published in 2019, a valid collaboration between gynecologist oncologists and hepatobiliary surgeons is shown, demonstrating that the presence of liver metastases should not preclude an attempt at SCS on the basis of potential morbidity, reporting a 94.1% rate of complete cytoreduction. MIS could be considered an effective surgical approach in selected patients with resectable liver disease [74], even in cases of localized spleen recurrences [75,76].

Similarly, Chiappetta et al. achieved a 5 year survival rate of 65% in patients who underwent lung metastasectomy for gynecological tract cancers, reinforcing the notion of personalized surgery, which means tailoring the technique (VATS vs thoracotomy) and resection extension (wedge vs anatomical resection; size of safety margin; lymph node dissection and localization) to the type of tumor [77]. Collaboration with thoracic surgeons with the use of MI techniques may be crucial, even in cases of cardio-phrenic nodal recurrence (Figure 3) [72,78] or mediastinal masses [79].

The need to discuss the management of recurrent lesions on an individual basis in an interdisciplinary context is crucial to offer the best possible oncological and surgical approach.

## 4. Molecular Biology and Future Perspectives

Recent advances in the molecular characterization and natural history of OC have opened novel perspectives based on better prognoses and higher tumor sensitivity to platinum-based therapy in the case of alterations of BRCA genes or HRD. Gallotta et al. reported a better PFS after hepatic resection of ROC in BRCA-mutated patients, thus providing a molecularly-based line of evidence and helping the choice of therapeutic approach in this sometimes-disputed clinical setting [74].

Despite recent clinical advances, more comprehensive information on the molecular characteristics of tumors is a priority. The contribution of surgery is the possibility of obtaining pathological and molecular characterization of the surgical specimen, providing a complete study of the disease, from diagnosis to possible biological modification during its natural history.

Newer studies focus on the molecular features of tumors, providing background biological information and prompting hypotheses about therapeutic choices, or provide evidence for pre-existing hypotheses, suggesting potential therapeutic targets. Targeted therapies and immunotherapies have emerged as novel treatment strategies for ovarian cancer, which drives the management of ovarian cancer towards individualized treatments.

Further applications of individualized treatments, like radiomics and radiogenomics, may offer a noninvasive tool for the evaluation of the tumor, aiming at improving the prediction of patient outcome, optimal triage, and offering the best treatment options [80,81].

## 5. Conclusions

The role of MI-SCS has been widely investigated in the literature. It has shown advantages compared with the open approach in terms of surgical outcomes: less blood loss, a shorter hospital stay, and a consequently adequate onset of adjuvant therapy. Furthermore, MIS seems to be not inferior to laparotomy, even in terms of oncological outcomes.

However, we have to consider that, since all the studies about MI-SCS in the literature are retrospective, including small groups of patients and different patterns of recurrence, it may be difficult to draw unique conclusions. Furthermore, because of the high level of expertise needed for this kind of surgery, MI-SCS should be performed in high-volume oncological centers.

In the balance between survival benefit and surgery-related morbidity during the maximum cytoreductive surgical effort, it is essential to select the appropriate candidates who will benefit from surgery without worsening quality of life. Age, clinical conditions, previous surgeries or radiotherapies, BMI, and anesthesiologic conditions may affect the choice of the best surgical approach. Pattern of recurrence showed relevant survival differences according to the anatomic site of recurrent disease, showing that lymph-nodal recurrences have the best outcome, followed by peritoneal and parenchymal recurrences. Lastly, preoperative imaging with a precise description of the recurrence and its relationships with adjacent organs, as well as laparoscopy itself, contribute to optimal surgical planning.

Despite the challenging management of ROC, a consequence of evolution in surgery is that personalized therapy is now a reality. SCS, especially with minimally-invasive approaches, plays an extremely important role; on the one hand, in the treatment of the recurrent disease itself, by providing an increased PFS compared to chemotherapy alone, without compromising patients’ quality of life; and on the other, in the study of the natural history of the disease, by obtaining pathological and molecular characterization for individualized therapies.

## Figures and Tables

**Figure 1 cancers-15-04769-f001:**
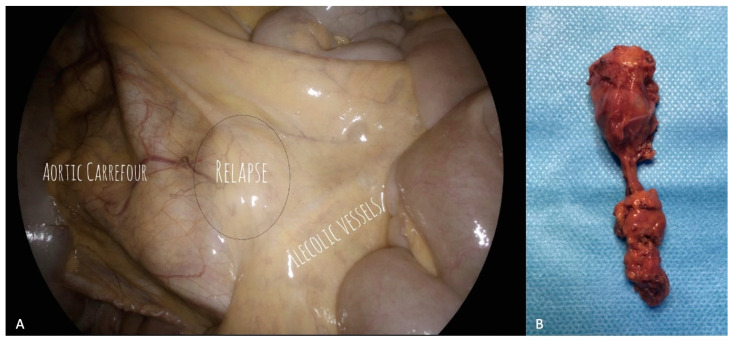
Para-caval lymph node recurrence. (**A**) Laparoscopic visualization of the recurrence and the surrounding field. The para-caval lymph node recurrence is clearly visible, just under the ileocolic vessels and above the aortic carrefour. (**B**) Surgical specimen. Histological examination confirmed the presence of metastasis of high grade serous ovarian cancer.

**Figure 2 cancers-15-04769-f002:**
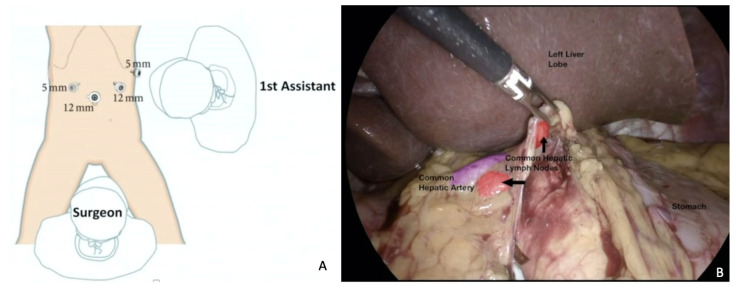
Superficial celiac isolated nodal recurrence successfully removed through a MI approach. (**A**) Trocar placement. Surgical planning is crucial and trocar placement is part of it. (**B**) Laparoscopic visualization of the recurrence. Here, a hepatic pedicle lymph node dissection is completed with control of anatomical structures, in particular the common hepatic artery (in purple). For a video of this procedure see ref [72].

**Figure 3 cancers-15-04769-f003:**
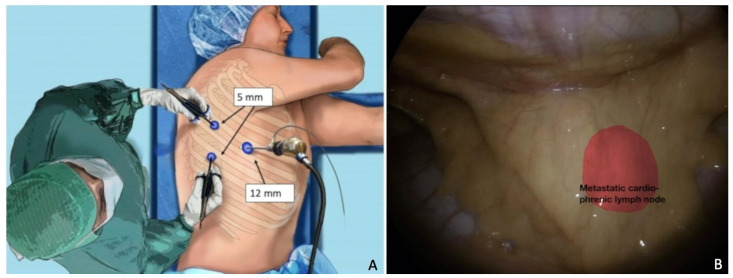
Cardio-phrenic nodal recurrence. (**A**) Trocar placement. The importance of surgical planning and the correct position of the patient. (**B**) Thoracoscopic visualization of the recurrence. In red, a metastatic cardio-phrenic lymph node. For a video of this procedure see ref [72].

## Data Availability

The data can be shared up on request.

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
