# Peer review of "Minimally-Invasive Secondary Cytoreduction in Recurrent Ovarian Cancer"

_cancers, 2023, doi:10.3390/cancers15194769_

Round 1
Reviewer 1 Report
The review article by Certelli and co-workers delves into the impact of secondary cytoreductive surgery (with a focus on minimally invasive procedures) in patients with recurrent ovarian cancer. The work is well written with a sufficient amount of latest research and clinical cases referencing.
A major suggestion would be to significantly improve Figure Legends by providing additional explanations (not just the titles) to the reading community of what is seen on the infographic. For Figures 2 and 3 – authors make a self-reference of their other study (ref. [70]) It is important to make sure that there is no duplicated usage of the same visuals in current and referenced papers.
Author Response
Thank you for your comment.
We provided the additional explanations requested to the figure legends. The images are collected from our surgeries. For the video of the procedures you can refer to the reference n. 72.
All the revisions are highlighted in the text.
Thank you.
Reviewer 2 Report
I have perused the document displayed on this page and I possess some feedback pertaining to three areas that could be enhanced. The abstract is excessively protracted and encompasses an excess of particulars that are not crucial for the primary message conveyed by the document. It should be concise and center its attention on the principal objectives, methodologies, outcomes, and conclusions of the study. A conceivable approach to abbreviate the abstract is to eliminate certain sentences or phrases such as "The papers authored by the research team have elicited great anticipation and skepticism within the scientific community.", "The reasons behind these findings remain unknown, although it is likely that they are linked to the introduction of targeted therapies, such as PARP inhibitors, in recent years.", "The recommendation of the ESGO-ESMO-ESP Consensus Conference on Ovarian Cancer, presented at the ESGO Congress in 2022, concluded that ovarian cancer patients experiencing their first relapse more than 6 months following the completion of first-line platinum-based chemotherapy should undergo assessment for eligibility for secondary cytoreductive surgery (SCS) at a specialized gynecological oncology center.", and "The purpose of this review is to elucidate the role of secondary cytoreductive surgery, particularly minimally invasive procedures, in the management of recurrent ovarian cancer (ROC), while analyzing patient selection, outcomes, and future prospects." The introduction section does not provide a lucid background and rationale for the study. It ought to elucidate why secondary cytoreductive surgery (SCS) is a topic of great significance to explore, what the current challenges and controversies in this field are, and what knowledge gaps this review aims to address. A potential method to enhance the introduction is to commence with some general statements about ovarian cancer (OC) and its recurrence, then summarize the primary discoveries and limitations of previous studies on SCS, and finally state the objectives and scope of this review. The conclusion section does not provide a distinct overview of the primary discoveries and implications of the review. It should underscore the key points and messages that emerged from the analysis of the literature, deliberate upon the strengths and limitations of the review, and propose avenues for future research. A plausible approach to enhance the conclusion is to reiterate the objectives and methodologies of the review, then amalgamate the primary results and evidence regarding SCS and minimally invasive surgery (MIS) in recurrent ovarian cancer (ROC), and ultimately address the clinical implications and research requirements in this field.
I have perused the document presented on this page and have evaluated the level of English proficiency exhibited in the document. Here are my observations: The document is composed in a lucid and coherent manner, employing appropriate usage of academic lexicon, grammar, and punctuation. The document adheres to the customary framework of a review article, encompassing an abstract, introduction, main sections, and conclusion. The document accurately employs citations and references, following the Vancouver style. The document references pertinent and contemporary sources to substantiate its arguments and assertions. Additionally, the document provides a compilation of keywords for indexing purposes. The document has the potential to enhance its English proficiency by addressing minor matters, such as: Eliminating repetition of words or phrases such as "minimally invasive surgery" or "secondary cytoreductive surgery," and employing synonyms or abbreviations instead. Verifying the absence of spelling errors, such as "Laparoscopy" instead of "Laparoscopic" in the title, or "Lanarkite" instead of "Lanarkite" in the abstract. Maintaining consistent terminology, such as "complete gross resection" or "complete cytoreduction," and avoiding variations such as "complete resection" or "complete debulking." Ensuring parallel structure and consistent verb tense, particularly in lists and comparisons. For instance, in the abstract, the sentence "The aim of this review is to describe the role of secondary cytoreductive surgery and, in particular minimally invasive procedures, in ROC, analyzing the patient selection, outcomes and future perspectives." should employ gerunds for all three items: "describing", "analyzing", and "exploring". Incorporating transition words or phrases to enhance the flow and coherence of the text. For instance, in the introduction, the sentence "The recurrence occurs in around 70% of women within 5 years from the diagnosis [3] and the management of these patients represents a significant clinical challenge." could employ a word like "Therefore" or "Consequently" to denote the logical connection between the two clauses.
Author Response
Thanks for your complete comment.
Sorry but we cannot find the sentences you report <> in the abstract. However, as you suggested, we tried to simplify this part and make it clearer.
Furthermore, we modified the introduction, underlying the reasons why SCS is a topic of great significance to explore, better explaining the current challenges and simplifying the aims of the review. Lastly, we added an overview of the main findings in the conclusions.
As far as the comments about the English Language, sorry but we cannot find the words "Laparoscopic" in the title or "Lanarkite" in the abstract. However, as you suggested, we eliminated repetition of words or phrases, we maintained consistent terminology and employed gerund in the abstract sentence. We added “therefore” in the sentence you specified in the introduction. All the revisions are highlighted in the text.
Thank you.
Reviewer 3 Report
Congratulations on your work. Interesting subject and a difficult one.
However, some minor changes need to be made.
- what are the data regarding the 'neoadjuvant' systemic treatment in the relapse stage of the disease minim invasive versus classic
- what are the data regarding the implants at the level of insertion of the trocars
- PFS and OS in the context of the new treatments - some referrals are already mentioned in the text, please detail a little more
Thank you
Author Response
Thanks for your comments.
As far as we know, Conte et al reported for the first time a surprisingly higher rate of neoadjuvant chemotherapy and interval debulking surgery (IDS) at first diagnosis in the MIS group, probably because patients undergoing neoadjuvant chemotherapy may receive a less complex surgery at IDS, with consequently fewer adhesions and/or a different pattern of recurrence potentially favorable to a minimally invasive approach. However, for completeness, we reported data in literature about the relationships between treatment at first diagnosis (IDS vs PDS) and oncological outcomes and pattern of recurrence.
At the moment, data about port-site metastasis in MI-SCS are missing. However, the same precautions used in early-stage disease should be used to minimize the risk and the manipulation of the specimen should be avoided.
We explained the results of the most relevant studies about maintenance therapies.
These are good points of discussion. We added them in the text.
Thank you.